# The Anti-ROR1 Monoclonal Antibody Zilovertamab Inhibits the Proliferation of Ovarian and Endometrial Cancer Cells

**DOI:** 10.3390/pharmaceutics14040837

**Published:** 2022-04-11

**Authors:** Dongli Liu, Gunnar F. Kaufmann, James B. Breitmeyer, Kristie-Ann Dickson, Deborah J. Marsh, Caroline E. Ford

**Affiliations:** 1Gynaecological Cancer Research Group, School of Clinical Medicine, Faculty of Medicine & Health, University of New South Wales, Sydney, NSW 2052, Australia; dongli.liu@unsw.edu.au; 2Oncternal Therapeutics, Inc., San Diego, CA 92130, USA; gkaufmann@oncternal.com (G.F.K.); jbreitmeyer@oncternal.com (J.B.B.); 3Translational Oncology Group, School of Life Sciences, Faculty of Science, University of Technology Sydney, Ultimo, NSW 2007, Australia; kristie-ann.dickson@uts.edu.au (K.-A.D.); deborah.marsh@uts.edu.au (D.J.M.); 4Northern Clinical School, Faculty of Medicine and Health, University of Sydney, Camperdown, NSW 2006, Australia

**Keywords:** ROR1, zilovertamab, cirmtuzumab, ovarian cancer, endometrial cancer

## Abstract

The non-canonical Wnt signalling receptor ROR1 is aberrantly expressed in numerous cancers, including ovarian and endometrial cancer. We previously reported that silencing ROR1 could inhibit the proliferation and metastatic potential of ovarian and endometrial cancer cells in vitro. Zilovertamab is an ROR1-targeting humanised monoclonal antibody, with demonstrated safety and efficacy in clinical trials of several ROR1-related malignancies. The aim of this study was to investigate the potential of zilovertamab alone, or in combination with commonly utilised gynaecological cancer therapies (cisplatin, paclitaxel and the PARP inhibitor—Olaparib) on high-grade serous ovarian cancer (HGSOC), including models of platinum resistance and homologous recombination deficiency (CaOV3, CaOV3CisR, PEO1 and PEO4) and endometrial cancer (EC) cell lines (Ishikawa and KLE). The effect of zilovertamab (at 25 µg/mL or 50 µg/mL) +/− agents was investigated using the IncuCyte S3 Live Cell imaging system. Zilovertamab alone inhibited the proliferation of HGSOC and EC cells in vitro, including in models of platinum resistance and homologous recombination deficiency. In general, the addition of commonly used chemotherapies to a fixed dose of zilovertamab did not enhance the observed anti-proliferative activity. This study supports the potential of zilovertamab, or other ROR1-targeting therapies, for treating women with HGSOC and EC.

## 1. Introduction

Expression of the non-canonical Wnt signalling receptor ROR1 is absent in most adult tissues, but it is aberrantly upregulated in a range of malignancies [1,2,3,4]. We previously reported that a high ROR1 level was associated with shorter survival in high-grade serous ovarian cancer (HGSOC) and endometrial cancer (EC) patients [5,6,7,8]. In addition, silencing ROR1 expression could inhibit the proliferation and metastatic potential of ovarian and EC cells in vitro [5,7,8,9].

Zilovertamab is a humanised monoclonal antibody against ROR1 that blocks Wnt5a-induced ROR1 signalling [10]. It has demonstrated safety and efficacy in Phase I/II clinical trials for a number of ROR1-expressing malignancies, including chronic lymphocytic leukemia (CLL) [11,12], mantle cell lymphoma (MCL) [13] and Her2-negative breast cancer (NCT02776917). In this study, we investigated the potential of the ROR1-targeting zilovertamab in inhibiting HGSOC and EC cell proliferation in vitro. For HGSOC, we selected ROR1-positive cell lines CaOV3, CaOV3CisR, PEO1 and PEO4 to model platinum-sensitive/resistant and homologous recombination (HR)-deficient/proficient ovarian cancer. Ishikawa and KLE were selected as moderate/high ROR1-expressing EC cell lines.

## 2. Methodologies

### 2.1. Cell Culture

High-grade serous ovarian cancer cell lines CaOV3, CaOV3CisR, PEO1 and PEO4, and endometrial cancer cell lines Ishikawa and KLE, were selected for positive ROR1 expression. CaOV3 (ATCC #HTB-75, Manassas, VA, USA) was a gift from Professor Anna DeFazio (Westmead Millennium Institute, Sydney, Australia). The cisplatin-resistant CaOV3 cell line (CaOV3CisR) was established by exposing CaOV3 cells to gradually increasing cisplatin concentrations as previously described [14]. PEO1 (ECACC #10032308, Porton Down, UK) and PEO4 (ECACC #10032309, Porton Down, UK) were gifts obtaining from Dr Simon Langdon (University of Edinburgh, Edinburgh, UK). Ishikawa (ECACC #99040201, Porton Down, UK) was generously gifted by Professor Jeff Holst (UNSW, Sydney, Australia), and KLE (ATCC #CRL-1622, Manassas, VA, USA) was kindly provided by Associate Professor Tracy O’Mara (QIMR, Brisbane, Australia). All cell lines were maintained in specific media (RPMI for ovarian cancer cell lines, MEM for Ishikawa and DMEM/F12 for KLE) supplemented with 10% foetal bovine serum (Scientifix, Melbourne, Australia), 1% GlutaMAX (ThermoFisher Scientific, Waltham, MA, USA) and 1% penicillin/streptomycin (ThermoFisher Scientific, Waltham, MA, USA), and subjected to routine mycoplasma testing.

### 2.2. Cell Viability Test

The half-maximal inhibitory concentration (IC50) for cisplatin (#S1166, Selleck Chemicals, Houston, TX, USA), paclitaxel (#S1150, Selleck Chemicals, Houston, TX, USA) and the PARP inhibitor Olaparib (S1060, Selleck Chemicals, Houston, TX, USA) in each cell line at 72 h was determined using the cell counting kit 8 (CCK-8, Sigma-Aldrich, Burlington, MA, USA). Zilovertamab was provided by Oncternal Therapeutics (San Diego, CA, USA). Cells were treated with a vehicle control or zilovertamab at 25 µg/mL or 50 µg/mL for 4 h prior to the addition of the chemotherapeutic agents at IC70 concentration. The effect of zilovertamab +/− agents was investigated using the IncuCyte S3 Live Cell Analysis System. Phase contrast cell images were obtained using a 10× objective lens within the instrument every 3 h for 72 h in total. The average confluence of each well was calculated and normalised against the baseline (time 0 after chemotherapy drugs were added).

### 2.3. qRT-PCR and Western Blot Analysis

Real-time reverse transcriptase PCR (qRT-PCR) was performed following the treatment, as previously described [7]. The relative expression levels of *ROR1, RHOA, VIM* (vimentin) and *CDH1* (E-cadherin) were calculated using the delta-delta Ct method and normalised against the mean of 3 reference genes (*SDHA*, *HSPCB* and *RPL13a*). The primers are listed in Appendix A. In addition, ROR1 (#AF2000, R&D Systems, Minneapolis, MN, USA) protein expression was analysed through a Western blot assay, as previously described [7].

### 2.4. Statistical Analysis

The IC50 and IC70 of individual drugs were estimated using non-linear regression dose–response (log inhibition vs. normalised response-variable slope) in GraphPad Prism (GraphPad Prism, San Diego, CA, USA). Differences in proliferation between different treatment arms were analysed using two-way ANOVA with a Dunnett correction for comparisons between the single treatment and the vehicle control, or a Tukey correction for comparisons between combined therapies and both single arms, for multiple *t*-testing with an adjusted *p*-value of 0.050 considered statistically significant. The additive model [15] was used to investigate the combination effect of zilovertamab and the chemotherapeutic agents. Briefly, the predicated viability of the drug combination was calculated as the product of cell viabilities of two drugs individually. The ratio of observed to predicted viabilities (also known as the survival index, or SI) indicates whether an interaction is additive (0.8–1.2), synergistic (<0.8) or antagonist (>1.2). The predicated SI of the drug combination in the additive model was defined as the product of the two single drugs. Statistical analysis was performed in GraphPad Prism 9 (GraphPad Prism, San Diego, CA, USA), with the significant cut-off set as *p* < 0.050.

## 3. Results

### 3.1. Single-Agent Zilovertamab Treatment Significantly Inhibits HGSOC Cell Proliferation

To investigate the effect of zilovertamab on the proliferation of HGSOC cells, we used the IncuCyte S3 live-cell imaging platform to monitor real-time cell confluency of the ROR1-positive HGSOC cell lines over a total period of 72 h following treatments (Figure 1A). At 72 h, zilovertamab alone at either 25 or 50 µg/mL significantly reduced the proliferation of CaOV3, CaOV3CisR and PEO1; the 50 µg/mL dose inhibited PEO4 proliferation (Figure 1B, Appendix A). Zilovertamab (50 µg/mL) alone significantly downregulated ROR1 expression levels in CaOV3 at the transcriptional level (Figure 2A). None of the changes in ROR1 protein passed the significance cut-off (0.05 Wilcoxon signed-ranks test) following single zilovertamab treatment in any of the cell lines (representative images are shown in Figure 2B). We also analysed the change in genes encoding epithelial–mesenchymal transition (EMT) markers (CDH1 and VIM, which encode E-cadherin and vimentin, respectively) and ROR1 initiated Wnt signalling pathway marker (RHOA). Zilovertamab treatment resulted in the marked upregulation of E-cadherin at transcriptional but not translational level in the CaOV3 cell line (Figure 2A,B). Compared with E-cadherin, the mesenchymal marker vimentin was expressed at a much lower level in CaOV3 and was increased at the transcriptional level following zilovertamab treatment. No significant changes in the EMT marker were observed following monotherapy of zilovertamab in other HGSOC cell lines.

In addition, we included adjuvant chemotherapies (cisplatin, paclitaxel and the poly(ADP)-ribose polymerase inhibitor (PARP) inhibitor (PARPi) Olaparib, commonly adopted in gynaecological cancer treatment. For each drug, the IC70 dose values 72 h post-treatment were calculated via fitting non-linear dose–response curves (Table 1, Appendix A). Single chemotherapy at IC70 dose tended to inhibit cell proliferation in individual cell lines (Appendix A); however, some of the effects were not significant after adjusting for multiple comparisons at 72 h (Figure 1B). Combining zilovertamab and other therapies did not significantly further inhibit cell proliferation compared with both single treatments at 72 h, except for zilovertamab (50 µg/mL) plus cisplatin in the PEO4 cell line (Figure 1B, Appendix A).

### 3.2. Single-Agent Zilovertamab Treatment Significantly Inhibited Endometrial Cancer Cell Proliferation

As seen in HGSOC cells, zilovertamab alone inhibited the proliferation of Ishikawa and KLE EC cell lines over the 72 h period (Figure 3A) and led to a significantly reduced cell confluency at 72 h (Figure 3B, Appendix A). Single chemotherapies at IC70 dose significantly inhibited the cell proliferation of Ishikawa and KLE at 72 h (Figure 3B). None of the combined treatments showed a superior effect on inhibiting cell proliferation to each single treatment in either Ishikawa or KLE (Figure 3B, Appendix A). No significant changes in ROR1 expression level at either transcriptional (Figure 4A) or translational levels (Figure 4B) were observed following monotherapy of zilovertamab.

### 3.3. Synergistic Effect of Combined Zilovertamab and Paclitaxel in Platinum-Resistant HGSOC

To investigate the combination effect of zilovertamab and other commonly used chemotherapeutic agents, we calculated the observed to predicted survival ratios (Table 2) based on the additive model [15], where numbers under 0.8 indicate synergistic, 0.8 to 1.2 indicate additive and values over 1.2 indicate antagonistic effects. The combination of zilovertamab (25 µg/mL) and Olaparib in the platinum-resistant CaOV3CisR cell line showed an additive effect, with the 95% confidence interval located in the 0.8 to 1.2 range (Table 2). However, the majority of effects observed by combining zilovertamab and chemotherapies in other HGSOC or EC cell lines were antagonistic.

## 4. Discussion

In this study, single-agent zilovertamab significantly inhibited the proliferation of HGSOC cells over 72 h, including in models of platinum resistance and homologous recombination deficiency (HRD), and in EC cells in vitro. A limit of 72 h was selected as the detection window based on the doubling time of the cell lines and half-life of all the therapies assessed in this study, as previously reported in [17]. Zilovertamab presented a long plasma half-life of 32.4 days in a recent Phase I clinical trial [12]; therefore, an expansion of the treatment period could be conducted in future single zilovertamab studies. No significant changes in the ROR1 expression level were observed following the zilovertamab treatment, aligning with previous reports that zilovertamab (previously known as cirmtuzumab) blocks the ROR1-signalling-induced activation of Rho-GTPases, including Rac1, RhoA and cdc42 [18,19], without changing ROR1 levels. We did not observe the regulation of RhoA at the transcriptional level following single zilovertamab treatment. The effect of zilovertamab on the active form of these GTPases warrants further investigation. The Cui et al. study previously reported that silencing ROR1 reversed the EMT procedure in breast cancer, as indicated by reduced SNAIL-1/2, ZEB1 and vimentin, and increased ZO-1 and E-cadherin [20]. EMT has been consistently implicated as a contributor to cancer metastasis [21,22] and has been significantly correlated with prognosis in ovarian and endometrial cancer patients [23,24]. In the current study, no significant EMT regulation at the transcriptional level following zilovertamab treatment in most of HGSOC or EC cells was observed (Figure 1C and Figure 2C). Instead, incubating CaOV3 with 50 µg/mL zilovertamab upregulated EMT at the translational level (lowered *CDH1* and increased *VIM,*
Figure 2A). The effect of zilovertamab in cell migration and cancer metastasis appears to be complex in HGSOC and EC.

The addition of zilovertamab did not further enhance the anti-proliferative effect of commonly used chemotherapies in most of the HGSOC and EC cell lines included in this study. Most of the combined therapies presented an additive or even antagonist effect. Zilovertamab appeared to be more effective as a monotherapy than in combination with common chemotherapy agents for HGSOC or endometrial cancer. However, it is important to note that the fixed doses of zilovertamab used in this study were based on previous studies in breast and ovarian cancers [25,26]. In breast cancer, the combination of zilovertamab and paclitaxel has been demonstrated to be more effective than either of the treatments alone in vivo [18], and formed the preclinical evidence for combined treatment in the HER2-negative breast cancer clinical trial (NCT02776917). The dose–response of zilovertamab is likely to vary between tumour types and cell lines. In our study, a higher dose (50 µg/mL) of zilovertamab was required to significantly inhibit cell growth in KLE, which had the highest ROR1 expression level of all the cell lines investigated. Further experiments investigating the combined effects of zilovertamab and other agents are needed, in order to direct combination therapy in ovarian and endometrial cancer.

In addition to the first-line chemotherapy agents cisplatin and paclitaxel, PARPis have shown clinical benefit in treating recurrent HGSOC patients, especially those presenting with HRD [27,28,29]. Compared with PEO4, zilovertamab at either 25 or 50 µg/mL induced more cell death in PEO1 (Appendix A). Our results suggest the potential of single zilovertamab therapy in treating both HR-deficient and HR-proficient HGSOC, but with no significant benefit observed from combined therapies.

This pilot study has provided the first preclinical evidence of the ROR1-targeting drug—zilovertamab—in HGSOC and EC in the context of the vast majority of subtypes (platinum-resistant/sensitive and HR-proficient/deficient, as well as endometrioid/high-grade serous). Our preliminary results support single-agent zilovertamab in future clinical trials in HGSOC and EC, and the potential of ROR1-taregeting therapies in treating HGSOC and EC. We recognise that a weakness of our study is the limited doses applied for zilovertamab. Previous clinical trials for zilovertamab selected the dose of zilovertamab considering the ROR1 surface level on circulating tumour cells. Future studies investigating correlations between ROR1 levels and dose responses to zilovertamab are warranted. In addition, transcriptional and translational regulation in ROR1-initiated Wnt signalling, EMT pathways and others followed by the effective doses (e.g., required amount to saturate ROR1 surface molecules on cells) of zilovertamab should be investigated in the future. To incorporate all the treatment conditions (single zilovertamab, chemotherapies and combined therapies), a general vehicle control (0.1% DMSO) was applied. However, because zilovertamab was developed as a monoclonal antibody, an IgG control could be included in future study to elicit nonspecific effect of the antibody.

## Figures and Tables

**Figure 1 pharmaceutics-14-00837-f001:**
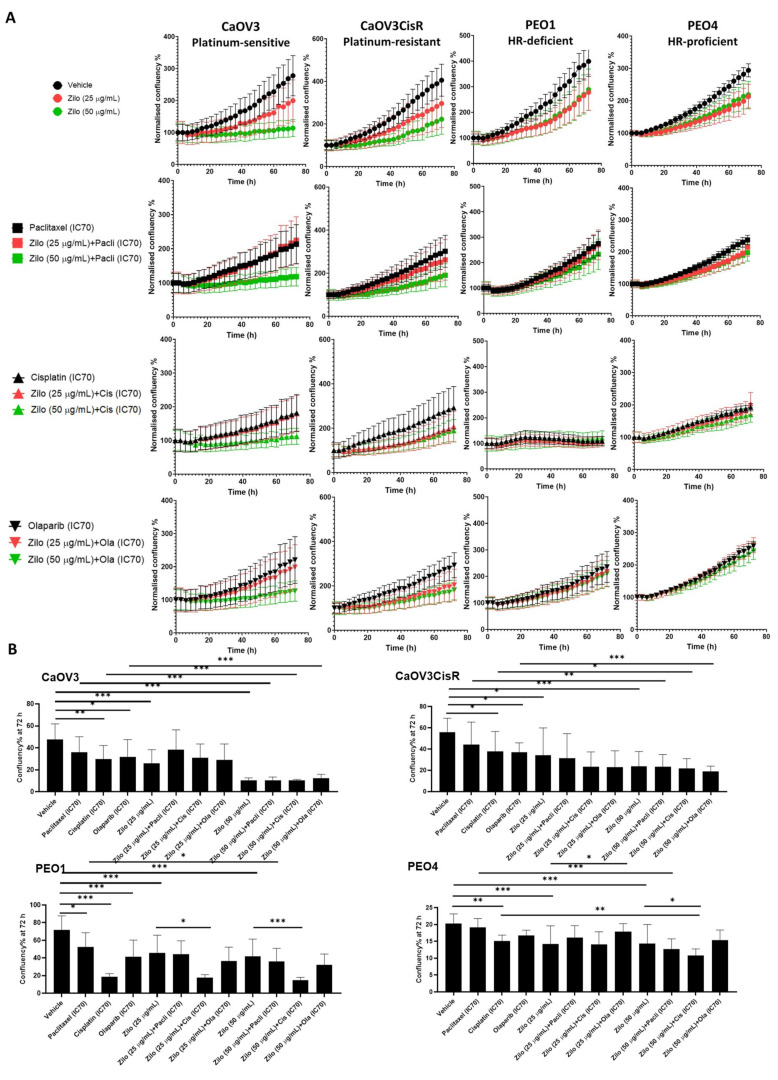
The effect of zilovertamab (“zilo”) +/− Paclitaxel, Cisplatin, Olaparib at IC70 doses on high-grade serous ovarian cancer cell lines CaOV3, CaOV3CisR, PEO1 and PEO4. (**A**). Confluency of cell lines over 72 h of treatments analysed by IncuCyte S3. The confluency of each time point was normalised against the baseline. (**B**). Differences in cell confluency following each treatment compared with the control (vehicle for single treatment, single arm for combined therapies) at 72 h. Significance of the comparisons was assessed using two-way ANOVA with a Dunnett/Tukey correction for multiple comparison. * Adjusted *p* < 0.05 ** Adjusted *p* < 0.01, *** Adjusted *p* < 0.001. For each panel, *n* = 5, error bar = SEM.

**Figure 2 pharmaceutics-14-00837-f002:**
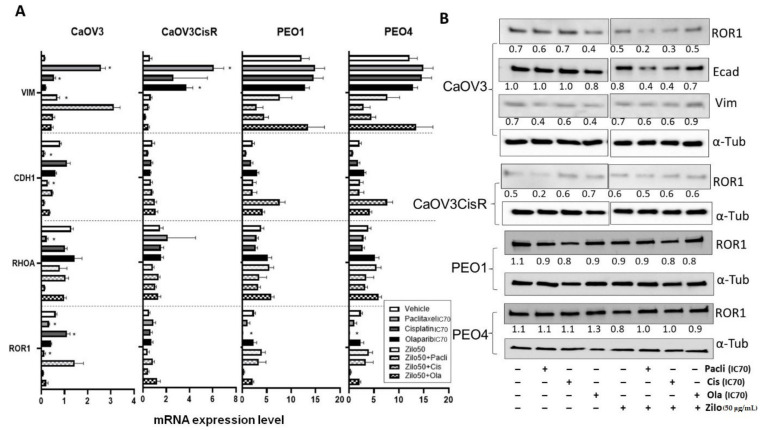
Real-time reverse transcription PCR and Western blot analysis following zilovertamab (“zilo”) +/− Paclitaxel, Cisplatin, Olaparib treatment in high-grade serous ovarian cancer cell lines CaOV3, CaOV3CisR, PEO1 and PEO4. (**A**). The relative mRNA level of ROR1, CDH1, RHOA and VIM following 72 h treatment. * Significant at *p* < 0.05 compared with vehicle. (**B**). Protein level of ROR1 following 72 h treatment as measured by Western blot (representative image). The numbers below each lane represent ratios of band intensity of protein versus α-Tubulin, calculated with ImageJ [16].

**Figure 3 pharmaceutics-14-00837-f003:**
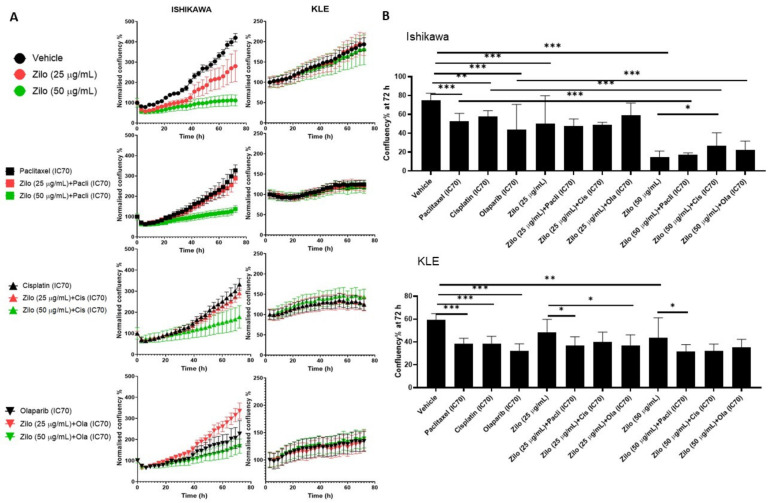
The effect of zilovertamab (“zilo”) +/− Paclitaxel, Cisplatin, Olaparib at IC70 on endometrial cancer cell lines Ishikawa and KLE. (**A**). Confluency of cell lines over 72 h of treatments analysed by IncuCyte S3. (**B**). Differences in cell confluency following each treatment compared with control (vehicle for single treatment, single arm for combined therapies) at 72 h. Significance of the comparisons was assessed using two-way ANOVA with a Dunnett/Tukey correction for multiple comparisons. * Adjusted *p* < 0.05 ** Adjusted *p* < 0.01, *** Adjusted *p* < 0.001. For each panel, *n* = 5, error bar = SEM.

**Figure 4 pharmaceutics-14-00837-f004:**
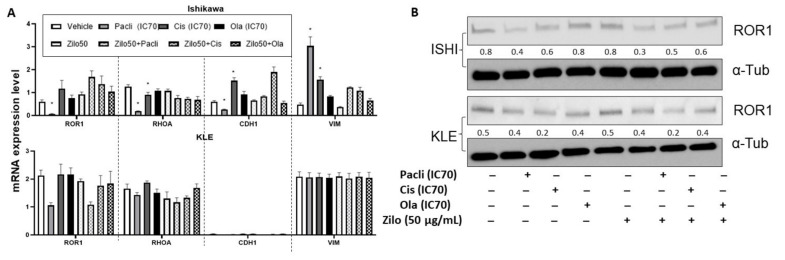
Real-time reverse transcription PCR and Western blot analysis following zilovertamab (“zilo”) +/− Paclitaxel, Cisplatin, Olaparib treatment in endometrial cancer cell lines Ishikawa and KLE. (**A**). The relative mRNA level of ROR1, RHOA, CDH1 and VIM following the treatments indicated by the qRT-PCR result. * Significant at *p* < 0.05 compared with the vehicle. (**B**). Protein level of ROR1 following the treatments indicated by Western blot. The numbers below each lane represent ratios of band intensity of protein versus α-Tubulin calculated with ImageJ [16].

**Table 1 pharmaceutics-14-00837-t001:** IC70 doses of reagents (paclitaxel, cisplatin and Olaparib) used for the high-grade serous ovarian cancer cell lines CaOV3, CaOV3CisR, PEO1 and PEO4, and endometrial cancer cell lines Ishikawa and KLE.

	CaOV3	CaOV3CisR	PEO1	PEO4	Ishikawa	KLE
Paclitaxel (nM)	0.788	0.649	3.620	3.038	6.852	12.680
Cisplatin (µM)	1.305	4.038	0.496	10.400	1.937	1.517
Olaparib (µM)	6.980	3.353	0.395	0.406	1.535	10.730

**Table 2 pharmaceutics-14-00837-t002:** Analysis of interaction between zilovertamab at 25 and 50 µg/mL doses and standard chemotherapy drugs at IC70 doses (as shown in Table 1) in high-grade serous ovarian cancer and endometrial cancer cell lines at 72 h. Survival index (SI) ratio listed in the table indicated synergistic (<0.8), subadditive (>1.2) or additive (0.8–1.2, grey shaded) effect of the drug combination. Data are presented as the mean (lower 95% confidence interval, upper 95% confidence interval).

	Zilovertamab (25 µg/mL)			Zilovertamab (50 µg/mL)		
	+Cisplatin	+Paclitaxel	+Olaparib	+Cisplatin	+Paclitaxel	+Olaparib
CaOV3	1.37 (1.18, 1.56)	1.47 (1.04, 1.90)	1.27 (1.10, 1.44)	1.51 (1.33, 1.69)	1.37 (0.92, 1.82)	1.41 (1.18, 1.64)
CaOV3CisR	1.10 (0.74, 1.46)	1.30 (0.74, 1.86)	1.01 (0.80, 1.22)	1.54 (0.00, 4.62)	1.36 (0.87, 1.85)	1.42 (0.08, 2.76)
PEO1	1.56 (0.95, 2.17)	1.57 (1.00, 2.14)	1.55 (0.81, 2.29)	1.74 (1.14, 2.34)	1.28 (0.84, 1.72)	1.40 (0.91, 1.88)
PEO4	1.72 (0.48, 1.96)	1.35 (0.90, 1.8)	1.56 (0.82, 2.30)	1.44 (0.47, 2.41)	1.28 (0.69, 1.87)	1.57 (0.26, 2.88)
Ishikawa	1.54 (0.96, 2.12)	1.63 (0.72, 2.54)	3.77 (0.45, 7.09)	2.02 (1.42, 2.62)	1.84 (0.77, 2.9)	3.73 (1.08, 6.38)
KLE	1.03 (0.81, 1.25)	0.99 (0.73, 1.25)	1.04 (0.64, 1.44)	1.40 (0.83, 1.97)	1.15 (0.57, 1.73)	1.29 (0.58, 2)

## Data Availability

Data is contained within the article and Appendix A associated with the article.

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
