# Peer review of "The Anti-ROR1 Monoclonal Antibody Zilovertamab Inhibits the Proliferation of Ovarian and Endometrial Cancer Cells"

_pharmaceutics, 2022, doi:10.3390/pharmaceutics14040837_

Round 1
Reviewer 1 Report
This is a very well-written manuscript that uses clear, concise language. It is entirely an in vitro investigation and should be reviewed as such. I feel that the authors should introduce to the methods section considerations related to Table 3, in particular how the survival index is determined. It should also discuss the cut-offs. Along these lines, a major finding is that Zilovertamab is effective alone as determined by the ratios in Table 3. However, the reader should have some estimate of the variability in the ratios and this should be added with significance testing. The Table 3 heading should list concentrations for cisplatin, paclitaxel, and olaparib. If a control antibody was or was not tested this should be discussed. The 72 hr duration of treatment should receive some expansion. Do the authors have any information on the half-life of the treatments in the cell culture environment? This should be discussed. In addition, they should discuss how a 72 hr exposure time is to be related to any in vivo extrapolations from the in vitro experimentation.
While I was able to understand Figure 1, I was not able to read all the labeling. It might be advisable to present panels A, B, & C as separate figures. Figure 2 suffers from the same and it is recommended that some remedy be applied. Bottom line: I enjoyed reading this work as a reviewer and believe it will have a readership when it is received modifications outlined here.
The strengths and weaknesses of the work should be included in the discussion.
Author Response
Response to Reviewer 1 comments:
- Point 1: I feel that the authors should introduce to the methods section considerations related to Table 3, in particular how the survival index is determined. It should also discuss the cut-offs.
- Response 1: To clarify this point, we have included more details on how we calculated the survival index in the “Statistical analysis” session of Methodologies as: “Briefly, the predicated viability of the drug combination was calculated as the product of cell viabilities of two drugs individually. The ratio of observed to predicted viabilities (also known as survival index, or (SI) indicates whether an interaction is additive (0.8 - 1.2), synergistic ( < 0.8) or antagonist ( > 1.2)”.
- Point 2: Along these lines, a major finding is that Zilovertamab is effective alone as determined by the ratios in Table 3. However, the reader should have some estimate of the variability in the ratios and this should be added with significance testing.
- Response 2: We have included the uncertainty of the ratios with 95% confidence interval (CI) of the mean ratios in the updated Table 2. We used the range of 95% CI instead of mean value to indicate if the SI was synergistic, subadditive or additive. We acknowledge the comment of the significant test, however, we do not believe it is appropriate to perform a one-sample T test for the means to check statistical significance for the nonparametric data (n=5).
- Point 3: The Table 3 heading should list concentrations for cisplatin, paclitaxel, and olaparib.
- Response 3: We appreciate the comments, but the IC70 does used for each drug varied in each cell line which makes it hard to incorporate in the heading of the table, we have referred to Table 1 in the updated heading of Table 2 (previous Table 3) to make it clear.
- Point 4: If a control antibody was or was not tested this should be discussed.
- Response 4: We did not include an isotype control antibody for zilovertamab for this study. The control we used in this study was vehicle (0.1% DMSO) as for all the other chemotherapy agents. We acknowledge the comment and have included additional discussion as: ”To incorporate all the treatment conditions (single zilovertamab, chemotherapies and combined therapies), a general vehicle control (0.1% DMSO) was applied. However, as zilovertamab was developed as a monoclonal antibody, an IgG control could be included in the future study to elicit nonspecific effect of the antibody.”
- Point 5: The 72 hr duration of treatment should receive some expansion. Do the authors have any information on the half-life of the treatments in the cell culture environment? This should be discussed. In addition, they should discuss how a 72 hr exposure time is to be related to any in vivo extrapolations from the in vitro experimentation.
- Response 5: According to the Phase I clinical trial in patients with chronic lymphocytic leukemia (CLL), zilovertamab has a long plasma half-life of 32.4 days (Choi et al., 2018). In this study, we selected 72h based on the mean doubling time (50 h) of the cell lines (CaOV3 68 h, PEO1 37 h, PEO4 59 h, Ishikawa 36 h). In addition, we would like to investigate if there was any combination effect of zilovertamab with other commonly applied chemotherapies or PARPi, thus we selected 72h as the detection window considering the half-life of all the drugs. The half-life for the common chemotherapies were 25- 49 min (initial phase) and 2-4 days (prolonged elimination phase) for cisplatin; 13 to 27 h in patients with metastatic breast cancer, or other solid tumours for paclitaxel; and 15 h for olaparib. 72h was also applied to calculate the IC75 dose of cisplatin used in high grade serous ovarian cancer cell lines in our recently published study (Supplementary Figure 1, Cole et al., 2021, Cellular & Molecular Life Sciences).
In regards to guiding future in-vivo studies, as our study was based on in vitro experiments purely which focused at the effect of the drug at cellular level, it remains limited in extrapolating in vivo metabolism of the drug. The real-time cell confluence monitored by the IncuCyte S3 system showed that at cellular level, single zilovertamab (50µg/ml) stared to induce significant (adjusted p<0.05) cell death within 72 h (CaOV3 30h, CaOV3CisR and PEO1 42h, PEO4 48h, Ishikawa 27h, KLE 51h). The completed Phase I clinical trial for zilovertamab in CLL showed the Plasma levels of cirmtuzumab
increased proportionally with dose, with peak concentrations exceeding 400 mg/mL for patients who received doses of 20 mg/kg.
We have included additional discussion in regards to the above points in the updated manuscript as: ”In this study, single agent zilovertamab significantly inhibited proliferation of HGSOC cells during 72 h, including in models of platinum resistance and homologous recombination deficiency (HRD), and in EC cells in vitro. A total of 72 h was selected as the detection window based on doubling time of the cell lines and half-life of all the therapies assessed in this study, as previously reported in [17]. As zilovertamab presented a long plasma half-life of 32.4 days in the recent Phase I clinical trial [12], an expansion of the treatment period could be conducted in future single zilovertamab studies.”
- Point 6: While I was able to understand Figure 1, I was not able to read all the labeling. It might be advisable to present panels A, B, & C as separate figures. Figure 2 suffers from the same and it is recommended that some remedy be applied.
- Response 6: We have updated Figure 1 and 2 as suggested and increased label font size to make them clearer.
- Point 7: The strengths and weaknesses of the work should be included in the discussion.
- Response 7: We have included additional discussions on strengths and limitations of our study in the updated version as: “This pilot study has provided the first preclinical evidence of the ROR1 targeting drug— zilovertamab in HGSOC and EC in the context of platinum resistant/sensitive and HR proficient/deficient as well as endometrioid/high grade serous models. Our preliminary results support single agent zilovertamab in future clinical trials in HGSOC and EC, and the potential of ROR1-targeting therapies in treating HGSOC and EC. We recognise that a weakness of our study is the limited doses applied for zilovertamab. Previous clinical trials for zilovertamab selected the dose of zilovertamab based on ROR1 surface level on circulating tumour cells. Future studies investigating correlation between ROR1 expression level and dose responses to zilovertamab is warranted. In addition, transcriptional and translational regulation in ROR1 initiated Wnt signalling, EMT pathways and others followed by the effective doses (amount to saturate ROR1 surface molecules on cells) of zilovertamab should be investigated in the future.”
Reviewer 2 Report
Comments are as attached here.

Author Response
Response to Reviewer 2 comments:
- Point 1: Figure 1 and 2, overall resolution is low. For 1A and 2A in particular, y-axis is invisible.
- Response 1: We appreciate this point of view, and have updated the figures with higher resolution. In the updated version, Figure 1 and 2 have been split into two figures each.
- Point 2: For drug zilovertamab (“zilo”), how 25 or 50 μg/ml is defined?
- Response 2: We selected the 50 μg/ml of zilovertamab based on previous in-vitro studies of zilovertamab (previously cirmtuzumab or UC-961) in ovarian cancer (Zhang et al., 2014) and breast cancer (Liu et al., 2015) from the lab that developed the antibody. The half dose (25 µg/ml) was applied to see if there was any effect following the increasing dose of zilovertamab. We have updated the corresponding session with an additional previous publication (Zhang et al., 2014) in Discussion as: “However, it is important to note that the fixed doses of zilovertamab used in this study were based on previous studies in breast and ovarian cancers”.
- Point 3: What are the doses for Paclitaxel, Cisplatin and Olaparib used?
- Response 3: We used IC70 doses of common chemotherapies (paclitaxel, cisplatin and olaparib) determined for each cell line for the combination analysis. Table 1 listed the detailed concentrations for IC70 doses. To make it clear, we included the dose information (IC70) of chemotherapies in the updated figure legends and labels.
- Point 4: The authors claim that “We also analysed the change in genes encoding epithelial-mesenchymal transition (EMT) markers (CDH1 and VIM) and ROR1 initiated Wnt signalling pathway marker (RHOA).” Any evidence from primary patient samples can validate their clinical significance, such as prognosis analysis?
- Response 4: We analysed RHOA gene to investigate if there was any transcriptional regulation in the ROR1 initiated Wnt signalling pathway following the treatments. CDH1 and VIM are genes encoding E-cadherin and vimentin which serve as key indicators of the epithelial-mesenchymal transition (EMT). EMT has been consistently implicated as a major contributor to cancer metastasis and have been associated with ovarian and endometrial cancers prognosis (Song et al., 2018; Ye et al., 2021). We have included the rationale in the updated Discussion as: “EMT has been consistently implicated as a contributor to cancer metastasis [19, 20] and has been significantly correlated with prognosis in ovarian and endometrial cancer patients [21, 22].”
- Point 5: Table 2 can be supplementary.
- Response 5: We have moved Table 2 into the supplementary document as suggested.
- Point 6: ROR1-positive cell lines CaOV3, CaOV3CisR, PEO1 and PEO4, before and after zilovertamab treatment, are ROR1 expression comparisons significant?
- Response 6: We did not observe significant differences in ROR1 mRNA before and after zilovertamab treatment except for CaOV3 (Figure 2A, Figure 4A). For ROR1 protein, no significant change was observed for any of the cell lines based on Wilcoxon signed-ranks test for the three times of Western blot. Figure 2B and 4B showed representative images of Western blot. This was consistent with previous in-vitro studies of zilovertamab (Zhang, et al. 2019. PNAS). We have clarified this in the updated version as:” Zilovertamab (50 µg/ml) alone significantly downregulated ROR1 expression levels in CaOV3 at the transcriptional level (Figure 2A). None of the changes in ROR1 protein passed the significance cutoff (0.05 from Wilcoxon signed-ranks test) following single zilovertamab treatment in any of the cell lines (representative images were shown in Figure 2B).”
Round 2
Reviewer 2 Report
No further comments.